# SSAL: Synergizing between Self-Training and Adversarial Learning for Domain Adaptive Object Detection

**Muhammad Akhtar Munir**[1*]**, Muhammad Haris Khan**[2]**, M. Saquib Sarfraz**[3,4]**, Mohsen Ali**[1]

{akhtar.munir@itu.edu.pk} {muhammad.haris@mbzuai.ac.ae}
{muhammad.sarfraz@kit.edu} {mohsen.ali@itu.edu.pk}

[1] Information Technology University of Punjab, [2] Mohamed bin Zayed University of Artificial Intelligence, [3] Karlsruhe Institute of Technology, [4] Daimler TSS

## Abstract

We study adapting trained object detectors to unseen domains manifesting significant variations of object appearance, viewpoints and backgrounds. Most current methods align domains by either using image or instance-level feature alignment in an adversarial fashion. This often suffers due to the presence of unwanted background and as such lacks class-specific alignment. A common remedy to promote class-level alignment is to use high confidence predictions on the unlabelled domain as pseudo labels. These high confidence predictions are often fallacious since the model is poorly calibrated under domain shift. In this paper, we propose to leverage model's predictive uncertainty to strike the right balance between adversarial feature alignment and class-level alignment. Specifically, we measure predictive uncertainty on class assignments and the bounding box predictions. Model predictions with low uncertainty are used to generate pseudo-labels for self-supervision, whereas the ones with higher uncertainty are used to generate tiles for an adversarial feature alignment stage. This synergy between tiling around the uncertain object regions and generating pseudo-labels from highly certain object regions allows us to capture both the image and instance level context during the model adaptation stage. We perform extensive experiments covering various domain shift scenarios. Our approach improves upon existing state-of-the-art methods with visible margins.

## 1 Introduction

Deep convolutional neural network based object detectors have shown promising results, through learning representative features from large annotated datasets [7, 32, 10]. However, like other supervised deep learning methods, object detection methods trained on the source domain do not generalize adequately to a new target domain. This problem, known as *domain shift* [49] could be exhibited by change in style, camera pose, or object size and orientation, or the number or location of objects in the scene, among other things. Often, collecting large annotated dataset for fine-tuning the model to the target domain is expensive, error prone and in many cases not possible. Unsupervised Domain Adaptation (UDA) is a promising research direction towards solving this problem by transferring knowledge from a labelled source domain to an unlabelled target domain.

---

*Corresponding author, Intelligent Machines Lab, Department of Computer Science, Information Technology University of the Punjab, Lahore, Pakistan. Email: akhtar.munir@itu.edu.pk Project Page: http://im.itu.edu.pk/synergizing-domain-adaptation/

35th Conference on Neural Information Processing Systems (NeurIPS 2021)

Many unsupervised domain adaptive detectors rely on adversarial adaptation or self-training techniques. Methods based on adversarial adaptation [4, 43, 15, 17, 54, 50, 3, 36], mostly rely on domain discriminator for aligning features at image or instance levels. However, due to the absence of labels in target domain they suffer from the challenges of how to pick samples for the adaptation. Selecting uniformly, one ends up missing on infrequent classes or instances. Most importantly adversarial alignment do not explicitly incorporates the class discriminative information, resulting in non-optimal alignment for classification and object detection tasks [43, 4, 45]. A potential solution to this problem is self-training based adaptation, however, it faces the challenge of how to avoid noisy pseudo-labels. Some methods choose high confidence predictions as pseudo-labels [27, 19, 42], but the likely poor calibration of model under domain shift renders this solution inefficient [38]. Further, in the case of object detection, prediction probability can not directly capture object localization inaccuracies.

We present a principled approach, dubbed as SSAL (Synergizing between Self-Training and Adversarial Learning for Domain Adaptive Object Detection), to achieve right balance between self-training and adversarial alignment for adaptive object detection via leveraging model's predictive uncertainty. To estimate predictive uncertainty of a detection, we propose taking into account variations in both the localization prediction and confidence prediction across Monte-Carlo dropout inferences [8]. Certain detections are taken as pseudo-labels for self-training, while uncertain ones are used to extract tiles (regions in image) for adversarial feature alignment. This synergy between adversarial alignment via tiling around the uncertain object regions and self-training with pseudo-labels from certain object regions lets us include instance-level context for effective adversarial alignment and improve feature discriminability for class-specific alignment. Since we select pseudo-labels with low uncertainty and take relatively uncertain as potential, object-like regions with context (i.e. tiles) for adversarial alignment, we tend to reduce the effect of poor calibration under domain shift, thereby improving model's generalization across domains.

Our key contributions include the following: (1) We introduce a new uncertainty-guided framework that strikes the right balance between self-training and adversarial feature alignment for adapting object detection methods. Both pseudo-labelling for self-training and tiling for adversarial alignment are impactful due to their simplicity, generality and ease of implementation. (2) We propose a method for estimating the object detection uncertainty via taking into account variations in both the localization prediction and confidence prediction across Monte-Carlo dropout inferences. (3) We show that, selecting pseudo-labels with low uncertainty and using relatively uncertain regions for adversarial alignment, it is possible to address the poor calibration caused by domain shift, and hence improve model's generalization across domains. (4) Unlike most of the previous methods, we build on computationally efficient one-stage anchor-less object detectors and achieve state-of-the-art results with notable margins across various adaptation scenarios.

## 2 Related Work

**Object detection.** Deep learning based object detection algorithms can be classified into either anchor-based [40, 30, 46, 2] or anchor-free methods [26, 6, 47]. Anchor-based methods, such as Faster RCNN [40], uses region proposal network (RPN) to generate proposals. Anchor-free detectors, on the other hand, skip proposal generation step and through leveraging fully convolutional network (FCN) [33] directly localize objects. For instance, [47] proposed per-pixel prediction and directly predicted the class and offset of the corresponding object at each location on the feature map. In this work, we capitalize on the computationally inexpensive characteristic in anchor-free detectors to study adapting trained object detectors.

**Tiling for object detection.** The process of cropping regions of an input image, a.k.a tiling, in a uniform [39], random, or informed [51, 16, 29] fashion before applying object detection is typically used to tackle scale variation problem and improve detection accuracy over small objects. Informed tiling can be achieved by first generating a set of regions of object clusters, and then cropping them for subsequent fine detection [51].

**Domain-adaptive object detection.** The pioneering work of [4] on domain-adaptive (DA) object detection proposed reducing domain shift at both image and instance levels via embedding adversarial feature adaptation into anchor-based detection pipeline. Global feature alignment could suffer as domains may manifest distinct scene layouts and complex object combinations. Several subsequent approaches attempted to achieve a right balance between the global and instance-level alignments

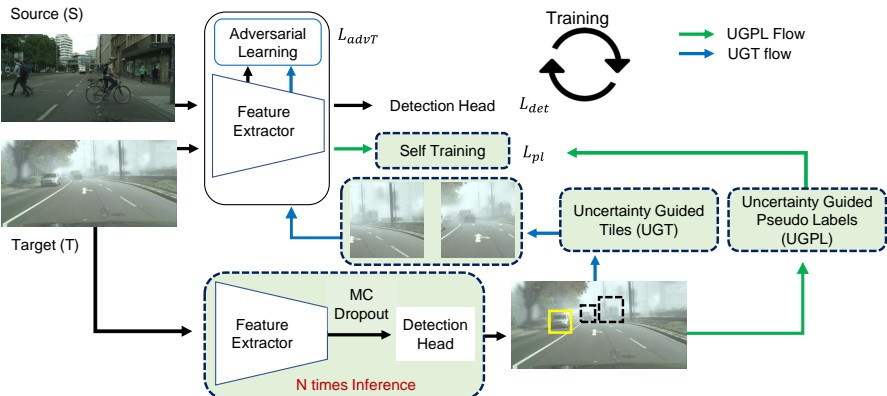

Figure 1: Overall architecture of our method (SSAL). Fundamentally, it is a one-stage detector [47] with an adversarial feature alignment stage. We propose uncertainty-guided self training with pseudo-labels (UGPL) and uncertainty-guided adversarial alignment via tiling (UGT) (in dotted boxes). UGPL produces accurate pseudo-labels in target image which are used in tandem with ground-truth labels in source image for training. UGT extracts tiles around possibly object-like regions in target image which are used with randomly extracted tiles around ground-truth labels in source domain for adversarial feature alignment.

[55, 50]. Other methods [15, 23, 1, 18] improved feature alignment in various ways e.g., through exploiting hierarchical feature learning in CNNs [15]. While above methods are built on two-stage pipeline, a few approaches have built domain adaptive detectors on one-stage pipeline [22, 17]. [17] proposed to predict pixel-wise objectness and center-aware feature alignment, building on [47], to focus on the discriminative parts of objects.

**Uncertainty for DA object detection.** Exploiting model's predictive uncertainty and entropy optimization have remained subject of interest in prior cross-domain recognition [34, 14, 35, 41] and detection [12, 37] works. For cross-domain recognition, [41] employed uncertainty for filtering training data and aligning features in Euclidean space. For DA object detection, [12] proposed an uncertainty metric to regulate the strength of adversarial learning for well-aligned and poorly-aligned samples adaptively.

**Pseudo-labelling for DA object detection.** In DA object detection, pseudo-labelling aims at acquiring pseudo instance-level annotations for incorporating discriminative information. Inoue et al. [19] generated pseudo instance-level annotations by choosing the top-1 confidence detections. Similarly, [42] obtained the same by using high-confidence detections and further refined them using tracker's output. Towards refining (noisy) pseudo instance-level annotations, [21] employed auxiliary component and [22] devised a criterion based on supporting RoIs.

Confidence-based pseudo-label selection is prone to generating noisy labels since the model is poorly calibrated under domain shift, eventually causing degenerate network re-training.

Unlike most prior methods we build on computationally inexpensive one-stage anchor-free detector. Different to existing methods, we leverage model's predictive uncertainty, considering variations in localization and confidence predictions across MC simulations, to achieve the best of both self-training and adversarial alignment through mining highly certain target detections as pseudo-labels and relatively uncertain ones as guides in the tiling process.

## 3 Proposed Method

In this section, we describe the technical details of our method. Fig. 1 displays the overall architecture of our method. We propose to leverage model's predictive uncertainty to strike the right balance between adversarial feature alignment and self-training. To this end, we introduce uncertainty-guided pseudo-labels selection (UGPL) for self-training and uncertainty-guided tiling (UGT) for adversarial alignment. The former allows generating accurate pseudo-labels to improve feature discriminability for class-specific alignment, while the latter enables extracting tiles on uncertain, object-like regions for effective domain alignment.

### 3.1 Preliminaries

**Problem Setting.** Let $\mathcal{D}_s = \{(x_i^s, \mathbf{y}_i^s)\}_{i=1}^{N_s}$ be the labeled source dataset and $\mathcal{D}_t = \{x_j^t\}_{j=1}^{N_t}$ be the unlabeled target dataset. Where $\mathbf{y}_i^s = \{\mathbf{b}_i^s, \mathbf{c}_i^s\}$ is set of bounding boxes $\mathbf{b}_i^s$ for the objects in the image $x_i^s$ and their corresponding classes $\mathbf{c}_i^s \in \{1, \ldots, C\}$. The source and target domains share an identical label space, however, violate the i.i.d. assumption since they are sampled from different data distributions. Our goal is to learn a domain-adaptive object detector, given labeled $\mathcal{D}_s$ and unlabeled $\mathcal{D}_t$, capable of performing accurately in the target domain.

**One-stage anchor-free object detection.** Owing to the computationally inexpensive feature of one-stage anchor-free detection pipelines, we build our uncertainty-guided domain-adaptive detector on fully convolutional one-stage object detector (FCOS) [47]. Inspired from the fully convolutional architecture [33], FCOS incorporates per-pixel predictions and directly regresses object location. Specifically, it outputs a $C$-dimensional classification vector, a 4D vector of bounding box coordinates, and a centerness score. The loss function for training FCOS is:

$$\mathcal{L}_{det}(\mathbf{c}_{u,v}, \mathbf{b}_{u,v}) = \frac{1}{N_{pos}} \sum_{u,v} \mathcal{L}_{cls}(\widehat{\mathbf{c}}_{u,v}, c_{u,v}) + \frac{1}{N_{pos}} \sum_{u,v} \mathbb{1}_{\widehat{c}_{u,v} > 0} \mathcal{L}_{box}(\widehat{\mathbf{b}}_{u,v}, \mathbf{b}_{u,v}) \tag{1}$$

where $\mathcal{L}_{cls}$ is the classification loss (i.e. focal loss [31], and $\mathcal{L}_{box}$ (i.e. IoU loss [52]) is the regression loss. $\widehat{\mathbf{c}}_{u,v}, \widehat{\mathbf{b}}_{u,v}$ denotes class and bounding box predictions at location $(u, v)$. $N_{pos}$ denotes the number of positive samples.

**Adversarial feature alignment.** Several methods [43, 4] align feature maps on the image-level to reduce domain shift via adversarial learning. It involves a global discriminator $D_{adv}$ that identifies whether the pixels on each feature map belong to the source or the target domain. Specifically, let $F \in \mathbb{R}^{H \times W \times K}$ be the $K$-dimensional feature map of spatial resolution $H \times W$ extracted from the feature backbone network. The output of $D_{adv}$ is a domain classification map of the same size as $F$. The discriminator can be optimized using binary cross-entropy loss:

$$\mathcal{L}_{adv}(x^s, x^t) = -\sum_{u,v} q \log(D_{adv}(F^s)_{u,v}), + (1-q) \log(1 - D_{adv}(F^t)_{u,v}) \tag{2}$$

where $q$ is the domain label $\in \{0, 1\}$. We perform adversarial feature alignment by applying gradient reversal layer (GRL) [9] to source $F^s$ and target $F^t$ feature maps, in which the sign of gradient is flipped when optimizing the feature extractor via GRL layer. Global alignment is prone to focusing on (unwanted) background pixels. We introduce uncertainty-guided tiling, that involves cropping tiles (regions with context) around object-like regions for effective adversarial alignment (sec. 3.2.1).

**Self-Training.** Self-training is a process of training with pseudo-labels, which are generated for unlabelled samples in the target domain with a model trained on labelled data. Hard pseudo instance-level labels are obtained directly from network class predictions. Let $\mathbf{p}_{j,k}$ be the probability outputs vector of a trained network corresponding to a detection $\widehat{\mathbf{y}}_{j,k}$, such that $p_{j,k}^c$ denotes the probability of class $c$ being present in the detection. With these probabilities, the pseudo-label can be generated for $\widehat{\mathbf{y}}_{j,k}$ as: $\widetilde{y}_{j,k}^c = \mathbb{1}[p_{j,k}^c \geq \alpha]$, where $\alpha = max_c p_{j,k}^c$. There could be a significant fraction of incorrectly pseudo-labelled detections used during training. A common strategy to reduce noise during training is to select pseudo-labels corresponding to high-confidence detections [19, 42]. Let $g_{j,k}$ be a boolean variable denoting the selection or rejection of $\widetilde{y}_{j,k}$ i.e. where $g_{j,k} = 1$ when $\widetilde{y}_{j,k}$ is selected or otherwise. Formally, in confidence-based selection, a pseudo-label $\widetilde{y}_{j,k}$ is selected as: $g_{j,k} = \mathbb{1}[p_{j,k}^c \geq \tau]$, where $\tau$ is the confidence threshold. These high confidence detections are often noisy because the model is poorly calibrated under domain shift. Instead, we propose to select pseudo-labels utilizing uncertainty in both class prediction and localization prediction to mitigate the impact of poor network calibration (sec. 3.2.1).

### 3.2 Uncertainty for Domain Adaptive Object Detection

The source model demonstrates poor calibration under target domain bearing sufficiently different superficial statistics and different object combinations [38, 45]. Although confidence-based selection (typically highest confidence) improves accuracy, the poor calibration of the model under domain shift

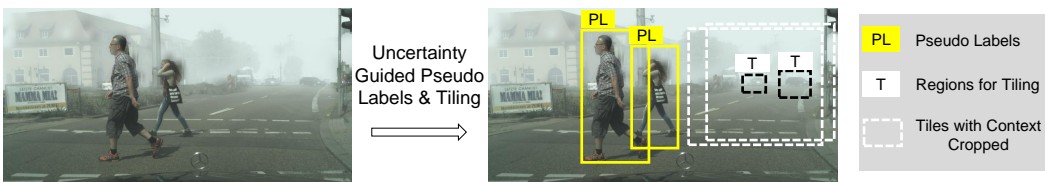

Figure 2: An illustration on which detections will be considered as pseudo-labels and which for extracting tiles. More certain detections, such as pedestrians are taken as pseudo-labels, whereas relatively uncertain ones, like cars under fog, are used for extracting tiles.

makes this strategy inefficient. As a result, it could lead to both poor pseudo-labelling accuracy and incorrect identification of possibly object-like regions for adversarial alignment. Since calibration can be considered as the model's overall prediction uncertainty [25], we believe that through leveraging model's predictive uncertainty we can negate the poor effects of calibration. To this end, we propose to leverage uncertainty in detections to select pseudo-labels for self-training and choose regions for tiling in adversarial alignment.

**Uncertainty in object detections.** Assuming one stage detector, we perform the uncertainty estimation by applying Monte-Carlo dropout [8] (in particular, spatial dropout [48]) to the convolutional filters after the feature extraction layer. Given an image $x$, we perform $N$ stochastic forward passes (inferences) using MC dropout. Let $\widehat{\mathbf{y}}_{n,m} = (\widehat{\mathbf{b}}_{n,m}, \widehat{c}_{n,m})$ be the $m_{th}$ detection in $n_{th}$ inference, $\widehat{c}_{n,m}$ be the class label with highest probability $\widehat{p}_{n,m}$ in the probability vector $\mathbf{p}_{n,m}$, and $\widehat{\mathbf{b}}_{n,m} \in \mathbb{R}^4$ is the predicted bounding box. We aim to capture the variations in both the localization prediction and confidence prediction across inferences. To this end, we define the uncertainty of the object detection prediction as mean class probability of the overlapping bounding boxes across individual inferences.

Specifically, for each $\widehat{\mathbf{y}}_{n,m}$, we create a set $\mathcal{T}_{n,m}$ by including all $\widehat{\mathbf{y}}_{k,l}$, where $k \neq n$ and $l$ is an arbitrary detection in $k_{th}$ MC forward pass, such that $\widehat{\mathbf{b}}_{n,m}$ has IoU with $\widehat{\mathbf{b}}_{k,l}$ greater than a specific threshold and $\widehat{c}_{n,m} = \widehat{c}_{k,l}$.

$$\mathcal{T}_{n,m} = \{\forall_{k \neq n} \cup (\widehat{\mathbf{b}}_{k,l}, \widehat{c}_{k,l}), \mid IoU(\widehat{\mathbf{b}}_{n,m}, \widehat{\mathbf{b}}_{k,l}) > \gamma , \ \widehat{c}_{k,l} = \widehat{c}_{n,m} \}. \tag{3}$$

Where $\gamma$ is the IoU threshold to identify bounding boxes occupying same region (detected as same object). We use $\mathcal{T}_{n,m}$ to estimate uncertainty based on both localization prediction and confidence prediction for $\widehat{\mathbf{y}}_{n,m}$ as:

$$\hat{p}_{n,m} = \frac{1}{|\mathcal{T}_{n,m}|} \sum_e \widehat{p}^e_{n,m}, \tag{4}$$

where $\widehat{p}^e_{n,m}$ is the class prediction confidence of $e_{th}$ detection in $\mathcal{T}_{n,m}$.

### 3.2.1 Uncertainty-Guided Pseudo-Labelling and Tiling

We interpret the averaged confidences $\hat{p}_{(.)}$ as a proxy (or indirect) measure of how uncertain (or certain) the model is in its class assignment and object localization information [41]. Under this definition, the model will be completely uncertain if $\hat{p}_{(.)}$ has uniform distribution whereas it will be completely certain if $\hat{p}_{(.)}$ can be represented by a Kronecker delta function.

**Uncertainty-guided pseudo-labelling for self-training.** As discussed above, the calibration can be considered as a measure of network's overall prediction uncertainty. To this end, we attempt to discover the relationship between calibration and individual detection uncertainties. We plot the relationship between the expected calibration error (ECE) score [13] and output detection uncertainties (Fig. 3). We see an existence of relationship between the ECE score and detection uncertainties. When we select pseudo-labels with more certain detections, the calibration error goes down significantly for this selected set. We hope that for this selected set of pseudo-labels, a high confidence detection will more likely result in a correct pseudo-label.

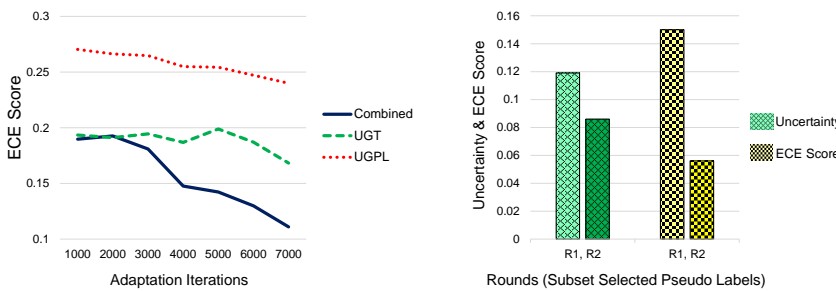

Figure 3: **Left.** ECE score as a function of UGT, UGPL, and our method that achieves synergy between UGT and UGPL, over the adaptation iterations. **Right.** Selecting more certain object detection pseudo-labels results in significant improvement in ECE score for this selected set over the adaptation course.

In the light of this observation, we propose to select the pseudo-label $\tilde{\mathbf{y}}_{j,k}$ corresponding to detection $\widehat{\mathbf{y}}_{j,k}$ by utilizing the uncertainty and detection consistency across $N$ inferences:

$$g_{j,k} = \mathbb{1}[\hat{p}_{j,k} \geq \kappa_1]\mathbb{1}[|\mathcal{T}_{j,k}| \geq \kappa_2], \tag{5}$$

where $\kappa_1$ and $\kappa_2$ are uncertainty and detection consistency thresholds. Fig. 2 illustrates some example detections that will be considered as pseudo-labels. Once the pseudo-labels are selected using Eq.(5), we use them to perform self-training as:

$$\mathcal{L}_{pl}(\tilde{c}_{u,v}, \tilde{\mathbf{b}}_{u,v}) = \frac{1}{N_{pos}} \sum_{u,v} \mathbb{1}_{\tilde{c}_{u,v}>0}\mathcal{L}_{cls}(\tilde{c}_{u,v}, \widehat{c}_{u,v}) + \frac{1}{N_{pos}} \sum_{u,v} \mathbb{1}_{\tilde{c}_{u,v}>0}\mathcal{L}_{box}(\tilde{\mathbf{b}}_{u,v}, \widehat{\mathbf{b}}_{u,v}) \tag{6}$$

where $\tilde{c}_{u,v}, \tilde{\mathbf{b}}_{u,v}$ represents the class label and bounding box coordinates of the (selected) pseudo-label. Compared to Eq. (1), in Eq. (6), we back-propagate classification loss only for (selected) pseudo-label locations.

**Uncertainty-guided tiling for adversarial alignment.** Existing image and instance-level adversarial feature alignment suffer from interfering background and noisy object localization. We propose uncertainty-guided tiling for adversarial alignment; it mines relatively uncertain detected regions, as possible object-like regions, for the tiling process. Tiling anchored by uncertain object regions allows adversarial alignment to focus on potential, however, uncertain object-like region with context (see Fig. 2). Specifically, if $g_{j,k} = 0$ corresponding to a detection $\widehat{\mathbf{y}}_{j,k}$ in Eq.(5), we consider it as an uncertain detection $\acute{\mathbf{y}}_{j,k}$ for extracting tile around it. Particularly, given $\acute{\mathbf{b}}_{j,k}$ as the bounding box for detection $\acute{\mathbf{y}}_{j,k}$, we crop a tile (region) $T_i$ of scale $W$ times as that of the detected bounding box. For source image, we randomly extract a tile $S_i$ around the ground-truth bounding box. After resizing both $T_i$ and $S_i$ to the input image size, we perform the adversarial alignment as:

$$\mathcal{L}_{advT}(S_i, T_i) = -\sum_{u,v} q \log(D_{advT}(F_T^s)_{u,v}) + (1-q)\log(1 - D_{advT}(F_T^t)_{u,v}), \tag{7}$$

where $F_T^s$ and $F_T^t$ are the feature maps for $S_i$ and $T_i$, respectively.

**Discussion.** We analyze the impact on model's calibration through the adaptation phase after (1) selecting pseudo-labels with more certain detections (UGPL), (2) performing tiling on relatively uncertain detections (UGT), and (3) achieving the the synergy between UGPL and UGT (our method). Model's calibration can be measured with Expected Calibration Error (ECE) score. We compute ECE score by considering both the confidence and the regression branch of the detector [24] [2]. Fig. 3 reveals that UGPL results in decreasing ECE score, and similarly (UGT) allows reducing the same even further. Finally, the synergy between UGPL and UGT achieves the lowest ECE score, significantly alleviating the impact of poor model's calibration under domain shift.

**Training objective.** We combine Eq.(1), Eq.(6), and Eq.(7) into a joint loss as $\mathcal{L} = \mathcal{L}_{det} + \mathcal{L}_{pl} + \mathcal{L}_{adv}$ and optimize it to adapt the source model to the target domain.

---

[2] Description on how ECE score is computed for detector is included in supplementary material.

| Method | person | rider | car | truck | bus | train | mbike | bicycle | mAP@0.5 | SO / Gain |
|---|---|---|---|---|---|---|---|---|---|---|
| **Two Stage Object Detector** | | | | | | | | | | |
| DAF [4] | 25.0 | 31.0 | 40.5 | 22.1 | 35.3 | 20.2 | 20.0 | 27.1 | 27.6 | 18.8 / 8.8 |
| SW-DA [43] | 29.9 | 42.3 | 43.5 | 24.5 | 36.2 | 32.6 | 30.0 | 35.3 | 34.3 | 20.3 / 14.0 |
| DAM [15] | 30.8 | 40.5 | 44.3 | 27.2 | 38.4 | 34.5 | 28.4 | 32.2 | 34.6 | 18.8 / 16.7 |
| CR-DA [50] | 32.9 | 43.8 | 49.2 | 27.2 | 45.1 | 36.4 | 30.3 | 34.6 | 37.4 | 22.0 / 15.4 |
| CF-DA [54] | **43.2** | 37.4 | 52.1 | **34.7** | 34.0 | **46.9** | 29.9 | 30.8 | 38.6 | 20.8 / 17.8 |
| HTCN [3] | 33.2 | 47.5 | 47.9 | 31.6 | **47.4** | 40.9 | 32.3 | 37.1 | 39.8 | 20.3 / 19.5 |
| UADA [36] | 34.2 | **48.9** | 52.4 | 30.3 | 42.7 | 46.0 | **33.2** | 36.2 | 40.5 | 20.3 / 20.2 |
| SAPNet [28] | 40.8 | 46.7 | **59.8** | 24.3 | 46.8 | 37.5 | 30.4 | **40.7** | **40.9** | 20.3 / **20.6** |
| **One Stage Object Detector** | | | | | | | | | | |
| Source Only | 31.7 | 31.7 | 34.6 | 5.9 | 20.3 | 2.5 | 10.6 | 25.8 | 20.4 | - |
| Baseline [17] | 38.7 | 36.1 | 53.1 | 21.9 | 35.4 | 25.7 | 20.6 | 33.9 | 33.2 | 18.4 / 14.8 |
| EPM [17] | 41.9 | 38.7 | 56.7 | 22.6 | 41.5 | **26.8** | 24.6 | 35.5 | 36.0 | 18.4 / 17.6 |
| **Ours (SSAL)** | **45.1** | **47.4** | **59.4** | 24.5 | **50.0** | 25.7 | **26.0** | **38.7** | **39.6** | 20.4 / **19.2** |
| Oracle | 47.4 | 40.8 | 66.8 | 27.2 | 48.2 | 32.4 | 31.2 | 38.3 | 41.5 | - |

Table 1: **Cityscapes → Foggy Cityscapes** Our method achieves an absolute gain of 19.2% over the source only model and out-performs most recent one-stage domain adaptive detector (EPM). SO refers to source only. The best results are bold-faced.

## 4  Experiments

**Datasets. Cityscapes** [5] dataset features images of road and street scenes and offers 2975 and 500 examples for training and validation, respectively. It comprises following categories: *person, rider, car, truck, bus, train, motorbike, and bicycle*.

**Foggy Cityscapes** [44] dataset is constructed using Cityscapes dataset by simulating foggy weather utilizing depth maps provided in Cityscapes with three levels of foggy weather.

**Sim10k** [20] dataset is a collection of synthesized images, comprising 10K images and their corresponding bounding box annotations.

**KITTI** [11] dataset bears resemblance to Cityscapes as it features images of road scenes with wide view of area, except that KITTI images were captured with a different camera setup. Following existing works, we consider car class for experiments when adapting from KITTI or Sim10k.

**Implementation Details.** FCOS [47], fully convolutional one-stage object detector, is trained over the source domain. During the adaptation process, using the source-trained model, we iterate over two steps: UGPL and UGT (Sec. (3.2.1)). Following [56, 57] we define going over these two steps once as *Domain Adaptation Round* or just *Round*. In all of the experiments for uniformity, we use three rounds. Since initially pseudo-labelling accuracy is likely poor, following [53], we perform adversarial domain adaptation (using UGT), in a round called $R0$. In next two rounds, $R1$ and $R2$, we apply both the self-training and adversarial domain adaptation using UGPL and UGT, respectively. For extracting tile around uncertain detection, a five times larger region is cropped around the center location. Height and width are re-adjusted to make the extracted tile square, so that during the resizing in any later stage the aspect ratio of any object in tile remains unaffected.

We use batch size of 3. Learning rate is set to $5 \times 10^{-3}$ during the training of source model and R0 round training, and then reduced to $1 \times 10^{-3}$ during the R1 and R2. $R1$ and $R2$ consists of $10K$ iterations, $R0$ however is consists of $5K$. IoU threshold $\gamma$ is set to 0.5. We use $N = 10$ MC-drop out inferences, with dropout rate set to 10%. All experiments are performed using a single GPU (Quadro RTX 6000). $\kappa_1$ and $\kappa_2$, uncertainty and detection consistency thresholds, are both set to 0.5, indicating object same class prediction and location should occur at-least 50% of times. All training and testing images are resized such that their shorter side has 800 pixels.

### 4.1  Comparison with state-of-the-art

For all the domain adaptation experiments we compare both existing state-of-the-art, one-stage and two-stage object detectors using the same feature backbone. Results are compared in terms of mAP(%), class-wise APs(%), and gain (%) achieved over a source only model. To better understand

|                     | Sim10K → CS        |              | KITTI → CS |             |
| ------------------- | ------------------ | ------------ | ---------- | ----------- |
| Method              | AP @ 0.5           | SO / Gain    | AP @ 0.5   | SO / Gain   |
| **Two Stage Object Detector** | | | | |
| DAF [4]             | 39.0               | 30.1 / 8.9   | 38.5       | 30.2 / 8.3  |
| SC-DA [55]          | 43.0               | 34.0 / 9.0   | **42.5**   | 37.4 / 5.1  |
| MAF [15]            | 41.1               | 30.1 / **11.0** | 41.0    | 30.2 / **10.8** |
| CF-DA [54]          | 43.8               | 35.0 / 8.8   | -          | -           |
| HTCN [3]            | 42.5               | 34.6 / 7.9   | -          | -           |
| SAPNet [28]         | **44.9**           | 34.6 / 10.3  | -          | -           |
| UADA [36]           | 42.0               | 34.6 / 7.4   | -          | -           |
| **One Stage Object Detector** | | | | |
| Source Only         | 38.0               | -            | 34.9       | -           |
| Baseline [17]       | 46.0               | 39.8 / 6.2   | 39.1       | 34.4 / 4.7  |
| EPM [17]            | 49.0               | 39.8 / 9.2   | 43.2       | 34.4 / 8.8  |
| **Ours (SSAL)**     | **51.8**           | 38.0 / **13.8** | **45.6** | 34.9 / **10.7** |
| Oracle              | 69.7               | -            | 69.7       | -           |

Table 2: **Sim10K → Cityscapes**: We outperform one-stage and two-stage object detectors both interms of mAP(%) and gain obtained over source. For this case, baseline value was recomputed. **KITTI → Cityscapes**: Our method outperforms both EPM and existing state-of-the-art methods with considerable margin in terms of mAP. SO refers to source only. The best results are boldfaced.

the effect of our algorithm, we also report results on **Baseline**, which is FCOS Tian et al. [47] along with global-level feature alignment. We discuss each experiment below.

**Weather Adaptation** (Cityscapes → Foggy Cityscapes)**.** Under same backbone and detection pipeline, our method outperforms the most recent one-stage domain adaptive detector (EPM) by an absolute margin of 3.6% and 1.6% in terms of mAP and gain. We report (Tab. 1) competitive performance against methods built on much stronger, two-stage anchor-based detection pipelines. In Fig. 5, compared to EPM [17], our method shows the capability of detecting objects of various sizes under severe climate changes.

**Synthetic-to-real** (Sim10K → Cityscapes)**.** Our method delivers a significant gain of 13.8% (Tab. 2). It exceeds existing state of the art methods, including ones built on stronger detection pipelines and feature backbones, by a notable margin, that is 2.8% mAP over top-performing one-stage adaptive detector (EPM) and 6.9% over two-stage object detection adaptation algorithm SAPNet [28].

**Cross-camera Adaptation** (KITTI → Cityscapes)**.** For this wide view camera setup to the normal scenario we achieve 45.6% mAP, as compared to results reported by the existing state-of-the-art algorithms using one-stage and two-stage detection pipelines, 43.2% and 42.5% (Tab. 2).

## 4.2 Ablation Studies

**Contribution of Components:** To analyze the effectiveness of each individual component in our proposed method we perform **Sim10K → Cityscapes** adaptation in different settings. Results are detailed in Tab. 3. We compare the impact on performance by training our model each time with (1.) confidence based pseudo labels only, obtained without our proposed uncertainty based selection. (2.) when only uncertainty-guided pseudo-labelling (UGPL) is used without the uncertainty-guided tiling procedure. and (3.) when relying only on uncertainty-guided tiling (UGT). Both UGPL and UGT show an increase of 11.5% & 12% in $AP@0.5$ over source only model and 3.5% & 4.0% over our Baseline. The non-trivial combination of UGPL and UGT, resulting in a synergy between them, produces a further 1.8% increase in $AP@0.5$ over their individual performance contributions. Especially in case of $AP@0.75$ our combined method reports 4.9 points improvement over the Baseline and more than 3 points improvement over UGPL and UGT, indicating that our method produces more accurate bounding boxes in the target domain.

**Impact of object sizes:** In Table 3, we also include the impact on performance of different components w.r.t object sizes. We use MS-COCO evaluation metric [32] to understand method's behavior with respect to different object sizes categorized as small (S):$< 32$ pixels, medium (M): between $32 - 96$ pixels and large (L): $> 96$ pixels.

**Uncertainty vs Confidence.** We contrast between the proposed uncertainty-guided balancing of pseudo-label (PL) selection and the tiling procedure and the confidence-guided balancing of these two procedures (Fig. 4(left)). Our approach resonates well with the fact that only when the model starts to become more certain of its detections, after round 1, the quantity of selected pseudo-labels should start to increase and so the number of regions being allocated to tiling should begin to decrease. This is not the case for the confidence based balancing. Through our adaptive allocation of detection

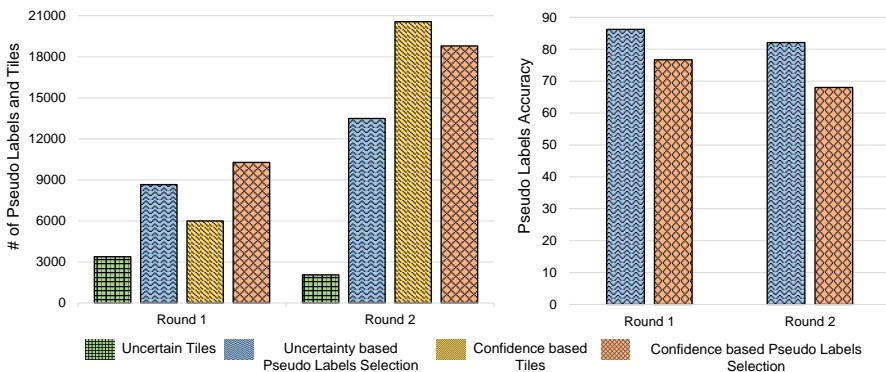

Figure 4: **Left.** Comparison of uncertainty-guided vs the confidence-guided selection of PL and tiles. **Right.** Low mean accuracy of confidence based selected PL vs the uncertainty based PL indicates uncertainty based PL selection is less noisy over the adaptation rounds. As the adaptation process progresses pseudo labels (for both type of selection) increases but uncertainty based PL remains less erroneous than confidence based PL.

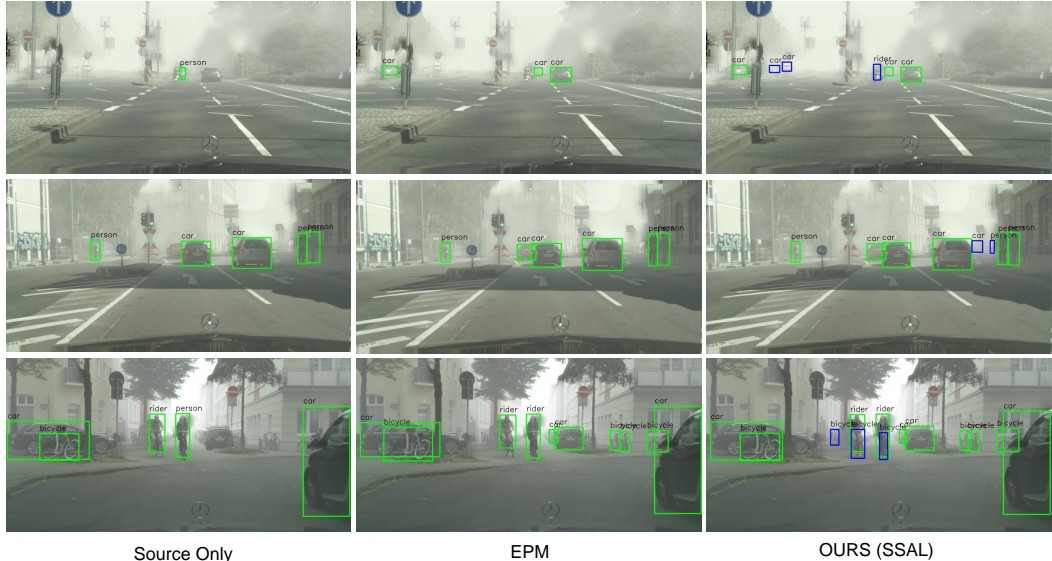

Figure 5: Detections missed by the EPM and found by our method are shown in Blue. Compared to EPM [17] our method achieves better adaptation.

regions, in Fig. 4(right) we demonstrate that our approach also delivers improved pseudo-labelling accuracy in both rounds compared to confidence-based selection.

**UGT vs Other Tile Selection Strategies.** We analyze the impact of extracting tiles centered around the uncertain detections (UGT) for adversarial learning in comparison to different tile selection strategies along with the Uncertainty Guided Pseudo Labels (UGPL) in Tab. 4. Specifically, we chose full image, random tiles, and certain tiles in adversarial learning with UGPL instead of proposed

| Methods | AP (mean) | AP @0.5 | AP @0.75 | AP @S | AP @M | AP @L |
|---|---|---|---|---|---|---|
| Source Only | 18.1 | 38.0 | 15.4 | 4.6 | 21.9 | 37.4 |
| Baseline | 25.9 | 46.0 | 25.5 | 5.7 | 28.8 | 52.2 |
| Confident PL | 21.8 | 43.2 | 19.8 | 4.7 | 27.5 | 42.9 |
| **Ours (UGPL)** | 27.6 | 49.5 | 26.9 | 6.7 | 31.2 | 55.0 |
| **Ours (UGT)** | 27.5 | 50.0 | 26.7 | 6.8 | 31.7 | 54.5 |
| **Ours (UGPL + UGT)** | **28.9** | **51.8** | **30.4** | 6.4 | **32.7** | **58.7** |

Table 3: Ablation results on **Sim10K → Cityscapes**. Combining the UGPL and UGT in a principled way results in most improvement than using them individually. Here, Baseline was recomputed by us.

| Combinations | AP@0.5 |
|---|---|
| Full Image + UGPL | 48.1 |
| UGPL | 49.5 |
| RandomTiles + UGPL | 49.8 |
| UGT | 50.0 |
| Certain Tiles + UGPL | 50.2 |
| UGT+UGPL | **51.8** |

Table 4: Comparison of proposed UGT vs other tiling strategies, including full image, random and certain tiles in the adversarial learning. We observe that compared to other tile selection strategies with UGPL, our proposed UGT provides maximum AP with UGPL.

(intelligent) tile selection process (UGT). Note that, when using random tiles there are various parameters (e.g.,location, size, and aspect ratio) involved in the tile selection process. So, we restrict the tile-selection space using the domain knowledge. Particularly, we restrict that the tile selected should have at least 60% of the image area. In case of certain tiles, tiling process is performed around the certain detections for the adversarial learning. We observe that compared to all three tile selection strategies with UGPL, our proposed UGT with UGPL provides maximum AP@0.5.

**Impact of R0.** To show how much R0 round contributes to the final performance, we report the performance of the base model (source only) after different rounds of adaptation for all three datasets adaptation scenarios. We report AP@0.5 after R0 and after R0+R1+R2 over the source model. As indicated in Tab. 5, performing both R1 and R2 rounds (that include both UGPL+UGT) results in significant improvement over when only R0 round (UGT) is performed.

| Datasets | Source Only | Source + R0 | Source+R0+R1+R2 |
|---|---|---|---|
| CS to Foggy CS | 20.4 | 27.4 | 39.6 |
| Sim10K to CS | 38.0 | 46.3 | 51.8 |
| KITTI to CS | 34.9 | 38.5 | 45.6 |

Table 5: Impact of R0 round. Performing both R1 and R2 rounds (UGPL +UGT) results in significant improvement over when only R0 round (UGT) is performed.

**Limitation.** Although we report improvement over the existing SOTA algorithms based on both one-stage and two-stage object detection pipelines, our method still faces challenges when dealing with small objects as depicted in Tab. 3. We plan to overcome this limitation by studying relationship between uncertainty, object sizes and related contexts.

## 5 Conclusion

We propose to leverage model's predictive uncertainty to achieve the best of self-training and adversarial learning for domain-adaptive object detection. Specifically, we propose to measure uncertainty in object detections by considering the variations in both the localization prediction and confidence prediction across Monte-Carlo dropout inferences. Certain detections are considered as pseudo-labels for self-training, while uncertain ones are used to extract tiles (regions in image) for adversarial feature alignment. This synergy between the both allows us incorporating instance-level context for effective adversarial alignment and improving feature discriminability for class-specific alignment. Further, it helps to reduce the effect of poor calibration under domain shift, thereby improving model's generalization across domains. Under various domain shift scenarios our method obtains notable improvements over the existing state-of-the-art methods.

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
