# Supplementary Material
# SSAL: Synergizing between Self-Training and Adversarial Learning for Domain Adaptive Object Detection

**Supplementary Material**

In this supplementary material, following sections are discussed: we include training algorithm (Sec. 1), analysis on the selection of drop out rate and hyperparameters used in our experiments (Sec. 2), ECE score calculation (Sec. 3), model calibration (Sec. 4) and more qualitative results (Sec. 5).

## 1  Algorithm

---

**Algorithm 1** SSAL: Training procedure with Uncertainty Guided Pseudo Labels ($UGPL$) and Uncertainty Guided Tiles ($UGT$)

---

**Input:** Set of labeled data, $\mathcal{D}_s$, and unlabeled data $\mathcal{D}_t$, uncertainty and detection consistency thresholds $\kappa_1 = 0.5$ & $\kappa_2 = N/2$ **Output:** Domain adapted trained model $G$

1: Train the model $G_s$, using labeled data, $\mathcal{D}_s$          ▷ Eq. (1)
2: **for** $i = 0$ to $\mathcal{R}$ **do**          ▷ Repeat until Completion of $\mathcal{R}$ Rounds
3:     $UGPL \leftarrow \phi$          ▷ empty set
4:     $UGT \leftarrow \phi$          ▷ empty set
5:     **if** $i == 0$ **then**
6:        UGT with $G_s$, $g_{j,k} = \mathbb{1}[\hat{p}_{j,k} < \kappa_1]\mathbb{1}[|\mathcal{T}_{j,k}| < \kappa_2]$ using Eq. (5) variant
7:        Train the model $G_i$, using UGT on $\mathcal{D}_t$ with $\mathcal{D}_s$          ▷ Eq. (1) & (7)
8:     **else if** $i \geq 1$ **then**
9:        UGPL with $G_{i-1}$, $g_{j,k} = \mathbb{1}[\hat{p}_{j,k} \geq \kappa_1]\mathbb{1}[|\mathcal{T}_{j,k}| \geq \kappa_2]$ using Eq. (5)
10:       UGT with $G_{i-1}$, $g_{j,k} = \mathbb{1}[\hat{p}_{j,k} < \kappa_1]\mathbb{1}[|\mathcal{T}_{j,k}| < \kappa_2]$ using Eq. (5) variant
11:       Train the model $G_i$, using UGPL and UGT on $\mathcal{D}_t$ with $\mathcal{D}_s$          ▷ Eq. (1), (6) & (7)
12:     **end if**
13:     $G \leftarrow G_i$
14: **end for**

---

## 2  Analysis

**On MC-dropout rate.** We show the impact on performance of our method with different dropout (spatial [6]) rates in Tab. 1. Our method mostly retains performance when perturbing the dropout rate from 10% to 30%. In particular, we see a maximum decrease of 0.8% in mAP score when increasing the dropout rate from 10% to 30%. This is expected as increasing the dropout rate increases prediction uncertainty which in turn affects the pseudo-label selection.

35th Conference on Neural Information Processing Systems (NeurIPS 2021)

| Dropout Rate | AP (mean) | AP @0.5 | AP @0.75 | AP @S | AP @M | AP @L |
|:---:|:---:|:---:|:---:|:---:|:---:|:---:|
| 30% | 28.2 | 49.4 | 27.5 | 5.9 | 31.1 | 58.1 |
| 20% | 28.1 | 50.3 | 28.0 | 6.1 | 32.5 | 56.0 |
| 10% | **28.9** | **51.8** | **30.4** | **6.4** | **32.7** | **58.7** |

Table 1: Impact on the performance of our method upon increasing dropout rates. We observe that our method is mainly robust against non-negligible variations in the dropout rates.

| $\kappa_1$ | AP (mean) | AP @0.5 | AP @0.75 | AP @S | AP @M | AP @L |
|:---:|:---:|:---:|:---:|:---:|:---:|:---:|
| 0.4 | 28.6 | 51.8 | 28.5 | 5.9 | 32.7 | 54.7 |
| 0.5 | **28.9** | **51.8** | **30.4** | **6.4** | **32.7** | **58.7** |
| 0.6 | 28.3 | 50.2 | 27.5 | 6.2 | 32.6 | 56.6 |

Table 2: Robustness of our method against variation in threshold hyperparameter $\kappa_1$, uncertainty threshold.

**On threshold hyperparameters.** We study the robustness of our method against variation in threshold hyperparameters $\kappa_1$ and $\gamma$ in Tab. 2 and Tab. 3, respectively. $\kappa_1$ is the uncertainty threshold and $\gamma$ is the IoU threshold. Although we set both thresholds at 0.5, we find that our method is relatively robust to these hyperparameters. For instance, upon varying the $\kappa_1$ by 0.1 unit in both directions, the maximum drop in mAP score is 0.6% (Tab. 2). In case of $\gamma$, we observe that IoU threshold = 0.5 gives stable results as compared to other values. Varying the $\gamma$ by 0.1 unit results into decreasing the performance over tight IoU thresholds.

| $\gamma$ | AP (mean) | AP @0.5 | AP @0.75 | AP @S | AP @M | AP @L |
|:---:|:---:|:---:|:---:|:---:|:---:|:---:|
| 0.5 | **28.9** | **51.8** | **30.4** | **6.4** | **32.7** | **58.7** |
| 0.6 | 28.3 | 49.8 | 28.5 | 5.9 | 32.4 | 58.2 |
| 0.7 | 27.5 | 50.4 | 27.9 | 5.4 | 31.5 | 55.8 |

Table 3: Robustness of our method against variation in threshold hyperparameter $\gamma$, IoU threshold.

## 3    ECE Score Computation

Our aim is to discover the relationship between (detection) model calibration and individual detection uncertainties. A standard measure for network calibration is expected calibration error (ECE) score [2, 7]:

$$ECE = \sum_{k=1}^{K} \frac{I(k)}{|\mathcal{D}|} | \sum_{x_i \in I(k)} max_c p_i^c - \sum_{x_i \in I(k)} \mathbb{1}[IoU(\widehat{\mathbf{b}}_i, \mathbf{b}_i) \geq 0.5] \mathbb{1}[\widehat{c}_i = c_i] |, \qquad (1)$$

where the confidence predictions on a dataset $\mathcal{D}$ (mostly testing set) are equally partitioned into $K$ bins. $I(k)$ is the number of examples falling in a specific bin k. To compute the calibration gap for each bin, the difference between the average accuracy and average confidence is computed. Note that we also take into account the regression branch output while computing accuracy [5]. The average over the calibration gap of all the bins results gives ECE score. In our case, we set $K = 10$ bins for ECE score computation.

## 4    Model's Calibration under Domain Shift

Tab. 4 reveals that a model trained on source domain (Sim10k [4]) suffers from poor calibration when tested on a target domain (Cityscapes [1]) manifesting distinct scene layouts and different object combinations. On the other hand, an oracle trained and tested on the target domain (Cityscapes [1]) shows significantly better calibration. Calibration is measured using ECE score.

| Models | ECE Score |
|---|---|
| Source Only | 0.25 |
| Oracle | 0.10 |

Table 4: Impact on (detection) model's calibration under domain shift. Calibration is measured using ECE score.

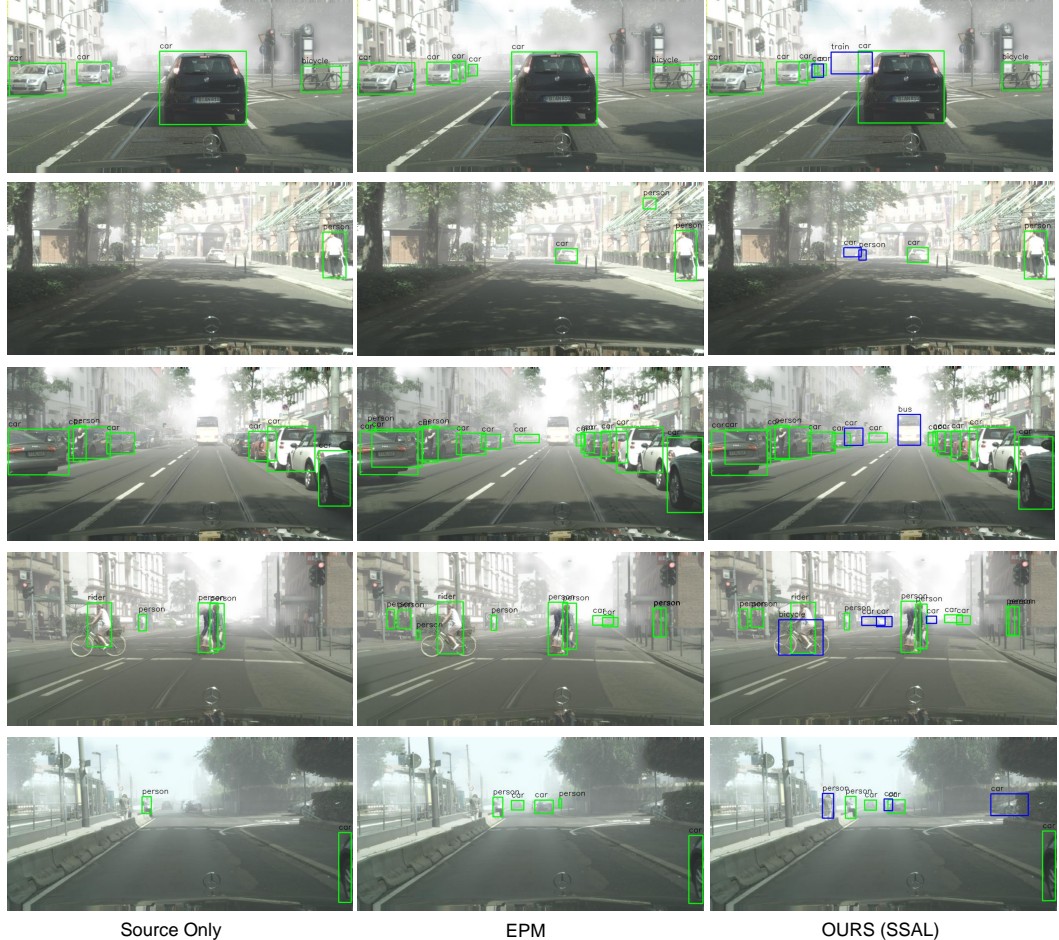

Source Only                    EPM                    OURS (SSAL)

Figure 1: More qualitative results. Detections missed by the EPM and found by our method are shown in Blue. Compared to EPM [3] our method is capable of detecting objects of various sizes under severe climate changes. Zoom-in for best viewing.

## 5 More Qualitative Results

Fig. 1 shows more qualitative results for source-only, EPM [3], and our method. We see that our method is capable of detecting objects at various scales under (severe) fog which are missed by EPM.