# OpenReview forum: "SSAL: Synergizing between Self-Training and Adversarial Learning for Domain Adaptive Object Detection"
_NeurIPS.cc/2021/Conference — NeurIPS 2021 Poster_

### Official Review · Reviewer_dBi3 · 2021-07-12

**Rating:** 4
**Confidence:** 4

**Summary:**

The work proposes a UDA pipeline for object detection based on self-training (via pseudo-labeling) and adversarial alignment. To this end, Monte-Carlo dropout inference is used to estimate detection confidence (or uncertainty). Certain detections (with  associated pseud-labels) are used for self-training while  uncertain ones are used to extract tile for alignment.



**Limitations And Societal Impact:**

I would argue that no discussion is needed on this topic. I cannot see any evident negative impact in adapting an object detector to a new domain.

**Main Review:**

1) The paper reads well and presents simple concepts, it is thus easy to understand. However, while the motivation behind using detections with low uncertainty for self-training is very intuitive, I could not grasp the reason behind exploiting low-confidence detections for adversarial aligment. I would accept it being verified experimentally, but no ablation in this direction is provided.

2) Novelty is quite limited, since all techniques employed are well known (MC- dropout, adversarial alignment, self-training via pseudo-labeling. Putting them together is somehow novel, but as I mention above, the motivation is unclear and it has not been proven nor verified experimentally.

3) Introduction, line 61: building over [47] cannot be claimed as a contribution

4) Related work, second paragraph can be omitted. Tiling is only marginally related to the work.

5) UGT paragraph (lines 202-211): here the authors try to motivate uncertainty-guided tiling as "Tiling anchored by uncertain object regions allows adversarial alignment to focus on potential, however, uncertain object-like region with context". Why not concentrating on those with high confidence or simply all of the detections. Since not motivation is brought forward, I would have expected an ablation study of this.

6) Ablation (especially table 3. and related paragraph - lines 268-280). Beside the missing ablation mentioned above, warmup seems to account for 2/3 of the total improvement over the baseline for mean AP (Baseline: 25.9, Ours w/o warmup: + 0.9%, Ours (UGPL + UGT)  + 2.9% )

7) Impact of object size (lines 281-284): no discussion on the (relatively) poor results for Small objects. Also you may want to discuss the reasons why warmup is detrimental for S(mall), almost irrelevant for M(edium) and essential for L(arge) objects.

Based on the points above, my overall score is below average. The missing ablation  (point 5) raises doubts about the soundness of the method, especially since there is no clear motivation for using only low confidence detections for adversarial alignment and being that one of the main claimed contributions.




**Time Spent Reviewing:**

2.5

---

> ### Author Response · Authors · 2021-08-10
> **Response to Reviewer dBi3**
>
>
> We would like to thank the reviewer for feedback and insightful comments. Following are the responses to the reviewer’s concerns.
>
> Comment #1:The paper reads well and presents simple concepts, it is thus easy to understand [...]
>
> Response#1: For the detections where the model’s prediction certainty is high, we use them as the pseudo-labels, that is Uncertainty Guided Pseudo Labels (UGPL), selection of less uncertain pseudo labels based on uncertainty estimation. However, where uncertain detection cannot be used for the pseudo-labels they might still contain the salient region. We exploit this to select tiles to perform feature-level alignment between source and target domain. This helps avoid information duplication and provides a principled way of selecting locations for the tile selection. Table 3 provides the ablation when we alone use Uncertainty Guided Tiling (UGT) for domain adaptive object detection, and in Figure 3 (left), ECE (Expected Calibration Error) score decreasing behavior also shows that the model is being calibrated over the iterations during training. Best performance is achieved when we combine the tiles and pseudo labels based on the model's predictive uncertainty.
> Please see Response #5 below for the ablation table and explanation.
>
> Comment #2:  Novelty is quite limited, since all techniques employed are well known [...]
>
> Response #2: Our main contribution is synergizing between self-training and adversarial alignment via leveraging uncertainty for object detectors under domain shift. It is a non-trivial approach of harnessing  MC (Monte Carlo) dropout based uncertainty estimation for intelligent tiling for adversarial alignment and self-training for effective domain adaptive object detection. Specifically, our proposed predictive uncertainty measure accounts for the overlap between predicted bounding boxes over MC-dropout simulations alongside class prediction probabilities, thus measuring uncertainty on all the parameters of accurate object detection. Strong empirical results under challenging domain shift scenarios corroborate the effectiveness of our method. Finally, we show that this synergy allows countering the poor calibration of the model under domain shift without actually using any post-hoc calibration method.
>
> Comment #5: UGT paragraph (lines 202-211): here the authors try to motivate uncertainty-guided tiling [...]
>
> Response #5: We performed the experiments taking into account different tile selection strategies, results are detailed below. As indicated by these results, using certain detections for the self-training (UGPL) and the tiles based on the uncertain detections for the adversarial feature alignment (UGT), results in the most accuracy.
> One reason could be that the uncertain detections are based on the regions where there is more feature misalignment across domains. Hence, performing adversarial feature alignment based on those regions is more intuitive.
>
>
> |                            | AP@0.5     |
> |:--------------------:        |:------:        |
> |   Full Image + UGPL      |  48.1          |
> |         UGPL                 |  49.5          |
> |  RandomTiles + UGPL      |  49.8          |
> |          UGT                 |  50.0          |
> | Certain Tiles + UGPL     |  50.2          |
> |       UGT+UGPL               |  51.8          |
>
>
> Full Image (adversarial learning) + UGPL:
> Upon using the full image in adversarial learning with UGPL, instead of the intelligent tile selection process (UGT), the performance deteriorates.
>
> RandomTiles (adversarial learning) + UGPL :
>
> When using random tiles with UGPL there are many parameters (location, size, aspect ratio) involved in the selection of the tiles. For the Random-tile selection in (Random Tile+UGPL) we use the domain knowledge to restrict the tile-selection space. Particularly, we restrict that the tile selected should have at least 60% of the image area. However, our method of selecting tiles around the uncertain detections still results in state-of-the-art results with considerable improvement.
>
> Certain Tiles (around certain detections for adversarial learning) + UGPL:
>
> Upon using certain tiles instead of uncertain tiles where we select UGPL as well, the experiment shows marginal improvement from state-of-the-art but still lower than our proposed method of using UGT with UGPL.
>
>
> Comment #6: Ablation (especially table 3. and related paragraph - lines 268-280) [...]
>
> Response #6: Line (238), we would like to reiterate that warmup (R0) round - part of our contribution - only does adversarial alignment based on uncertainty-guided tiling (UGT) to reduce error propagation in later rounds (R1 and R2). Such error propagation is likely if pseudo-labeling is allowed in the R0 round as the majority of them would be inaccurate at the beginning of adaptation. Table 3 validates the effectiveness of performing R0 round.
>
>
>
> Comment #7: Impact of object size (lines 281-284): no discussion on the (relatively) poor results for Small objects [...]
>
> Response #7: Since the ratio of small objects in the dataset is much smaller than the medium and large sized objects, the tiles are more likely to capture the large objects. Hence performing unsupervised adversarial learning for the feature alignments over such tiles, the model will be optimized to detect large objects better. This has been mentioned in the paper as a limitation, we hope to study this aspect more in future work.  However, please note that the AP drop for small objects is 0.5% (with R0 vs w/o R0), whereas AP improvement of large objects is 6.7%.
> Dataset distribution of small, medium, and large objects on Cityscapes in Sim10k to Cityscapes adaptation:
>
> Small: 5749
>
> Medium: 11569
>
> Large: 8862

---

> > ### Author Response · Authors · 2021-09-03
> > **Note to reviewer dBi3**
> >
> > As a final remark we want to state that a major concern of the reviewer was w.r.t to tiling ablations (justification of using uncertain tiles) and the impact of warmup. We have provided the additional tiling-choice ablation for the former and also have provided an ablation to distill the improvement from warmup (our round R0 with proposed UGT only) vs the (UGT+UGPL) in the response to reviewer pTSg. We reproduce the table clarifying warmup impact here again in case the reviewer haven't seen it in our other response.
> >
> > As indicated in the table below, performing both R1 and R2 rounds (UGPL +UGT) results in significant improvement over when only warmup R0 round (UGT) is performed.
> >
> > |			|Source Only  | Source + R0(warmup)| Source+R0(warmup)+R1+R2|
> > |----------------		|-------------	|-----------------------------|-----------------			    |
> > | CS to Foggy CS 	| 20.4    	| 27.4                    	 | 39.6        	|
> > | Sim10K to CS   	| 38.0    	| 46.3                    	 | 51.8        	|
> > | KITTI to CS		| 34.9    	| 38.5                    	 | 45.6        	|
> >
> >
> > We strongly feel that in our rebuttal response we have clarified all the concerns raised in points 1-7.
> >
> > We truly hope that the reviewer will consider these or at least provide an additional comment on the reasons where the reviewer still disagrees with our further ablations/clarifications in our response.

---

### Official Review · Reviewer_pTSg · 2021-07-16

**Rating:** 8
**Confidence:** 3

**Summary:**

 In this paper, authors propose an unsupervised domain adaptation method for object detection based on pseudo-labeling and adversarial training. The main idea is two-fold: 1) Monte-Carlo dropout is used to produce the detection uncertainty information that is used to generate pseudo-labels; and 2) regions surrounding uncertain detections are used for adversarial training against the regions from the source domain surrounding the ground truth bounding boxes. The proposed method is compared with the state-of-the-art single-stage and two-stage object detectors on Cityscapes-to-Foggy Cityscapes, KITTI-to-Cityscapes and Sim10K-to-Cityscapes datasets. Additionally, an ablation study examining the contribution of each of the proposed modules to the increased detection precision is presented.


**Limitations And Societal Impact:**

The limitations regarding the decreased detection accuracy of the small objects is discussed in the paper. However, the proposed method involves multiple stages of training and multiple additional modules, which is why a discussion of how much additional memory / trainable parameters / more training time is needed for this method compared to the others is needed.


**Main Review:**

The paper is overall well-written, except the abstract which is a little bit hard to read.

The proposed idea of using spatial MC dropout to predict the detection and localization uncertainty is sufficiently novel and clever.  It is also intuitive to use the certain predictions as pseudo-labels. The only issue that I see is that the uncertain detections are used for adversarial training, as if they actually always correspond to the foreground objects. It is very much possible that some of the uncertain predictions are actually the false positives, and it is not clear to me how adversarially matching them to the source foreground objects would benefit the detection accuracy.

The experimental design and the choice of baseline methods as well as an ablation study provide conclusive and convincing evidence that both of the proposed methods lead to increased detection precision of medium and large objects.

While Figure 3 clearly illustrates that both the uncertainty and calibration error are minimized by the proposed solutions, the message behind the Figure 4 is unclear to me. It is difficult to understand what the authors were trying to show with these histograms, so perhaps a more detailed explanation in the caption with a brief conclusion would make it easier to comprehend.


**Time Spent Reviewing:**

3

---

> ### Author Response · Authors · 2021-08-10
> **Response to Reviewer pTSg**
>
> We would like to thank the reviewer for feedback and insightful comments. We appreciate the reviewer’s positive comments regarding the extensive analysis, experiments, and ablations. Following are the responses to the reviewer’s questions.
>
> Comment #2: The proposed idea of using spatial MC dropout to predict the detection and localization uncertainty is sufficiently novel and clever [...]
>
> Response #2: We agree with the reviewer that being an unsupervised detection in the target domain, the regions selected for tiling may contain false positives. We want to highlight some statistics related to the target domain dataset that tells us that our uncertainty guided tiling actually incorporates more foreground regions as compared to the background. It makes intuitive sense since although uncertain, these are still object detection predictions of the model. We computed below some statistics on Sim10k to Cityscapes adaptation scenario to provide some insights. R0, R1, R2 are training rounds.
>
>
> R0    216 tiles were based on false positive region in target domain out of 5171 tiles
>
> R1    124 tiles were based on false positive region in target domain out of 3386 tiles
>
> R2    105 tiles were based on false positive region in target domain out of 2062 tiles
>
>
> Comment #3: While Figure 3 clearly illustrates that both the uncertainty and calibration error are minimized by the proposed solutions [...]
>
> Response #3: As suggested, we have updated the captions of Fig. 3 and 4, and lines of draft (285-292) to make them easier to comprehend. Briefly we explain it below
> Fig.4 We are comparing confidence based pseudo-label selection vs uncertainty based one. As indicated on the right side, the uncertainty based pseudo-label selection is more accurate, even when the number of certainty based pseudo-label has increased (Left).

---

> > ### Comment · Reviewer_pTSg · 2021-08-18
> > **Concerns regarding the use of uncertain tiles are addressed, but more ablation study is needed.**
> >
> > After reading the feedback from fellow reviewers, as well as the author's response, I still think this paper presents a decent work that should be accepted to the venue with minor modifications.
> >
> > Our common concern regarding the choice of uncertain tiles for the adversarial training was cleared by the authors in the rebuttal with a tiling choice ablation study and with the statistics of the number of false positives chosen for the adversarial training. These results must be included and properly discussed in the main paper.
> >
> > With all due respect, the argument of novelty that was brought up by the fellow reviewers, is largely subjective, unless the main idea has literally nothing new in it, which is clearly not the case. Even if all the logical parts introduced in the paper are not necessarily novel, the authors presented a way of combining them to get the inductive bias needed for solving the task.
> >
> > I tend to share the concern of Reviewer dBi3 about the warmup. I think in order to exclude the possibility that the warmup contributes to most of the gain authors should present the results of the Baseline + warmup.

---

> > > ### Author Response · Authors · 2021-08-23
> > > **Response to Reviewer pTSg**
> > >
> > > Thanks for the comment, we will certainly include our tiling ablations and discussion in the paper.
> > >
> > > It seems that the term warmup, as used in our paper, has caused some confusion. We would like to take this opportunity to clarify it further.
> > >
> > > First, we would like to emphasize that warmup (R0 round) means that in the first round we only perform adversarial alignment based on our proposed Uncertainty-Guided Tiling (UGT) to reduce error propagation in the later (R1 and R2) rounds. Such error propagation is likely if pseudo-labeling is allowed in the first round as the majority of them would be inaccurate at the beginning of adaptation. In Table 3 (in the main paper) we only wanted to show the effectiveness of performing the first adaptation round (R0) without including our uncertainty based pseudo-labels (UGPL).
> > >
> > > As suggested, to clarify the impact of the warm up (Reviewer dbi3) we have performed some more experiments.
> > >
> > > To show how much R0 (warmup) contributes to the final performance we report the performance of the base model (source only) after different rounds for all three datasets adaptation scenarios. We report AP@0.5 after R0 and after R0+R1+R2 over the source model. As indicated in the table below, performing both R1 and R2 rounds (UGPL +UGT) results in significant improvement over when only warmup R0 round (UGT) is performed.
> > >
> > > |            |Source Only  | Source + R0(warmup)| Source+R0(warmup)+R1+R2|
> > > |----------------        |-------------    |-----------------------------|-----------------                |
> > > | CS to Foggy CS     | 20.4        | 27.4                         | 39.6            |
> > > | Sim10K to CS       | 38.0        | 46.3                         | 51.8            |
> > > | KITTI to CS        | 34.9        | 38.5                         | 45.6            |

---

### Official Review · Reviewer_eqrv · 2021-07-18

**Rating:** 6
**Confidence:** 3

**Summary:**

Authors address the issue of domain adaptation for object detection, to detect objects in images with same label space but drawn from different data distributions. To this end, the paper proposes to mine uncertain tiles including object-like regions for adversarial feature alignment, and mine certain pseudo-labels for self training. Authors argue that the uncertainty guided training strategy can lead to better calibration under domain shift and improve the model's generatlization. Experiments are carried out on street driving datasets and demonstrate the effectiveness of the proposed method.

**Limitations And Societal Impact:**

Authors have adequately addressed the limitations and potential negative societal impact of their work, or there are no obvious issues regarding that direction.

**Main Review:**

Mining pseudo-labels with low uncertainty for training makes sense to me and should lead to more robust training on the target domain images. However, it's not very clear why it would be optimal to select tiles with uncertain detections. It would be better to provide more experiments about the adversarial alignment part, especially about different sample selection strategy (random selection, keeping all detection outputs, etc).

In Table 1 and Table 2, although those two-stage models should have more powerful detection performance, they generally are worse in the source only settings. It may suggest that those baselines are not well-tuned or powerful enough. It would be interesting to see if the proposed method can also be applied to two-stage methods and provide consistent performance gains.

**Time Spent Reviewing:**

2.5

---

> ### Author Response · Authors · 2021-08-10
> **Response to Reviewer eqrv**
>
> We would like to thank the reviewer for feedback and insightful comments. We appreciate the reviewer’s positive comments regarding the mining of less uncertain pseudo labels. Following are the responses to the reviewer’s concerns.
>
> Comment #1: Mining pseudo-labels with low uncertainty for training makes sense to me [...]
>
> Response #1: More certain detections are used for pseudo-labels (bounding box and object class label). Using just these pseudo-labels the adaptation is slow. We extract the tiles centered around the uncertain detections and use those tiles (Uncertainty Guided Tiling (UGT)) for the adversarial learning. As suggested, we analyze the impact of different tile selection strategies along with the Uncertainty Guided Pseudo Labels (UGPL).
>
> |                            | AP@0.5     |
> |:--------------------:        |:------:        |
> |   Full Image + UGPL      |  48.1          |
> |         UGPL                 |  49.5          |
> |  RandomTiles + UGPL      |  49.8          |
> |          UGT                 |  50.0          |
> | Certain Tiles + UGPL     |  50.2          |
> |       UGT+UGPL               |  51.8          |
>
> Full Image (adversarial learning) + UGPL:
> Upon using the full image in adversarial learning with UGPL, instead of the intelligent tile selection process (UGT), the performance deteriorates.
>
> RandomTiles (adversarial learning) + UGPL    :
>
> When using random tiles with UGPL  there are many parameters (location, size, aspect ratio) involved in the selection of the tiles. For the Random-tile selection in (Random Tile+UGPL) we use the domain knowledge to restrict the tile-selection space. Particularly, we restrict that the tile selected should have at least 60% of the image area. However, our method of selecting tiles around the uncertain detections still results in state-of-the-art results with considerable improvement.
>
>
>
> Certain Tiles (around certain detections for adversarial learning) + UGPL:
>
>  Upon using certain tiles instead of uncertain tiles where we select UGPL as well, the experiment shows marginal improvement from state-of-the-art but still lower than our proposed method of using UGT with UGPL.
>
>
>
>
>
> Comment #2: In Table 1 and Table 2, although those two-stage models should have more powerful detection performance [...]
>
> Response #2: The reviewer points out a good insight and we share a similar observation. We want to highlight that numbers reported in Table-1 and Table-2 (of two-stage in source only) were taken directly from the published works. One of the reasons could be that since two-stage methods are over-parameterized than the single-stage ones, they are more prone to overfitting on the source domain. In this study, although we build on the computationally efficient one-stage detector, our domain adaptive method is architecture-agnostic and will be equally effective with two-stage detectors. We plan to explore this in future work.

---

> > ### Comment · Reviewer_eqrv · 2021-09-03
> > **After reading the rebuttal**
> >
> > The rebuttal addresses part of my concerns and I appreciate the extra ablation results provided from the authors. I believe this paper is of interest to some audiences thus I am changing the score to 6.

---

### Official Review · Reviewer_WDr2 · 2021-07-20

**Rating:** 6
**Confidence:** 5

**Summary:**

This paper measures predictive uncertainty on class assignments and the bounding box predictions. The uncertainty is further used to guide the self-training and adversarial learning for domain adaptive object detection. The experiments cover various domain shift scenarios and prove the approach improves upon existing state-of-the-art methods.

**Limitations And Societal Impact:**

Yes

**Main Review:**

Overall, this paper is easy to follow. The experiments validate the effectiveness of the method.

1. However, my major concern about this paper is its novelty. Using uncertainty estimation to facilitate self-training and adversarial learning is a common operation in domain adaptation. The unique component of this paper is the way to measure uncertainty. As this is the core component of the method and the modifications are incremental, I hope to see more intuitions or conclusions that can help us guide the design of uncertainty estimation. More analysis about various uncertainty estimations are demanded to support the paper. While this paper only conducts a simple ablation study, which is not sufficient for the acceptance of the paper.

2. As the paper also talks about the complementary between self-training (A) and adversarial learning (B), is it possible to observe more improvements with A + B than independently using the two components? (A bring ) What’s the difference between this synergy with [22]?

Joint considering the effective experiments and the limited novelty, I give my ranking as borderline reject. I am open to hearing from the authors to clear the main contributions of their paper vs. related works.

Detailed comments:

There is only a Section 3.2.1 upon Section 3.2 which is not proper.


**Time Spent Reviewing:**

6

---

> ### Author Response · Authors · 2021-08-10
> **Response to Reviewer WDr2**
>
> We would like to thank the reviewer for feedback and insightful comments. We appreciate the reviewer’s positive comments regarding the effectiveness of our method. Following are the responses to the reviewer’s concerns.
>
> Comment #1: However, my major concern about this paper is [...]
>
>
> Response #1: To our knowledge, there is no domain adaptive object detector that aims to synergize between self-training and adversarial alignment using the model's predictive uncertainty. To this end, we also relate models’ predictive uncertainty with the domain shift problem in object detection and adapt existing classification based uncertainty estimation methods to object detection problems under domain shift. Specifically, our predictive uncertainty measure accounts for the overlap between predicted bounding boxes over MC (Monte Carlo) dropout based inferences alongside class prediction probabilities. Strong empirical results under challenging domain shift scenarios corroborate the effectiveness of our method. Further, we show that this synergy allows countering the poor calibration of the model under domain shift without actually using any post-hoc calibration method (please see Fig. 3). Finally, we also share that our method’s performance is agnostic to the specific choice of mc-dropout method commonly used in literature (table below).
>
> We show it on Sim10k to Cityscapes adaptation scenario.
>
> |                                   |     AP@0.5    |
> |-----------------------------------|---------------|
> |     UGT+UGPL (DropBlock)          |     47.1      |
> |     UGT+UGPL (Spatial Dropout)    |     51.8      |
>
>
> Comment #2: As the paper also talks about the complementary between self-training (A) and adversarial learning (B),  [...]
>
>
>
> Response #2: We have presented the individual component performance in comparison with a joint performance in our paper (Table. 3).
> As for difference with [22], most importantly, [22] did not leverage uncertainty estimation for self-training and adversarial learning. They incorporate positive and negative samples by weak self-training (on the basis of supporting RoI- classification probability) and background score regularization for adversarial training. In contrast, we leverage the model's predictive uncertainty based on IOU thresholds of multiple final detections using MC dropout that are potentially more accurate than the [22], for selecting certain pseudo labels and use uncertain detections to extract tiles for adversarial learning. Where [22] explicitly crafted different mechanisms to handle positives, negatives, and background (for self-training and adversarial learning), we on the other hand have proposed a unified method based on uncertainty to synergize between self-training and adversarial learning. This helps us in an adversarial alignment process that does not suffer from background clutter.
>
>
> We will also incorporate changes in the current manuscript as per the suggestions of the reviewer (organization of a section).

---

### Author Response · Authors · 2021-09-01
**Final note to all reviewers/AC**

Thanks for providing useful reviews. We would like to take this opportunity to add a final note. In our rebuttal response to each reviewer we have clarified the main issues and have provided the asked ablations on the tiling choice study (a common concern) as well as further clarified the impact of warmup with additional ablation (in response to reviewer PTSg and dBi3). In our responses we have also clarified the novelty aspect and re-iterated the core contributions of introducing uncertainty based synergy between self training and adversarial learning for effective domain adaptation of the model and showing its relevance in improving poor calibration of current object detection models under distribution shift. Thanks to reviewer PTSg for acknowledging and highlighting the novelty aspect of our contributions.

With the exception of reviewer PTSg, none of the other reviewers have engaged further in the rolling discussion period. We, therefore, sincerely hope that there was a discussion on our responses between the reviewers. In any case we hope that reviewers will consider our rebuttal responses and additional clarifying ablations before reaching a final score.

---

### Decision · Program_Chairs · 2021-09-27

**Decision:**

Accept (Poster)

**Comment:**

Following the rebuttal and discussion phase two reviewers increased their scores leading to final recommendations of one clear accept, two leaning towards accept and one rejection rating. After considering all reviews and discussion between authors and reviewers, the AC agrees with the reviewer consensus that this work merits acceptance. The key concern prior to the rebuttal was whether the combination of prior approaches (such as MC-dropout, adversarial alignment, and pseudo-label training) were sufficiently new. The rebuttal provided sufficient clarification on this point, especially through new ablation studies. The authors should make sure to include these ablation studies as well as additional text clarifying the novelty of their contribution in the final version.